# Multichannel Multiscale Two-Stage Convolutional Neural Network for the Detection and Localization of Myocardial Infarction Using Vectorcardiogram Signal

**Jay Karhade** [1], **Samit Kumar Ghosh** [1], **Pranjali Gajbhiye** [1], **Rajesh Kumar Tripathy** [1] and **U. Rajendra Acharya** [2,3,4,*]

1. Department of Electrical and Electronics Engineering, BITS-Pilani, Hyderabad Campus, Hyderabad 500078, India; f20180852@hyderabad.bits-pilani.ac.in (J.K.); samitnitrkl@gmail.com (S.K.G.); gajbhiyepranjali@gmail.com (P.G.); rajeshiitg13@gmail.com (R.K.T.)
2. School of Engineering, Ngee Ann Polytechnic, Singapore 599489, Singapore
3. Department of Bioinformatics and Medical Engineering, Asia University, Taichung 41354, Taiwan
4. School of Management and Enterprise, University of Southern Queensland, Springfield, Ipswich, QLD 4300, Australia
* Correspondence: Rajendra_Udyavara_ACHARYA@np.edu.sg

**Abstract:** Myocardial infarction (MI) occurs due to the decrease in the blood flow into one part of the heart, and it further causes damage to the heart muscle. The 12-channel electrocardiogram (ECG) has been widely used to detect and localize MI pathology in clinical studies. The vectorcardiogram (VCG) is a 3-channel recording system used to measure the heart's electrical activity in sagittal, transverse, and frontal planes. The VCG signals have advantages over the 12-channel ECG to localize posterior MI pathology. Detection and localization of MI using VCG signals are vital in clinical practice. This paper proposes a multi-channel multi-scale two-stage deep-learning-based approach to detect and localize MI using VCG signals. In the first stage, the multivariate variational mode decomposition (MVMD) decomposes the three-channel-based VCG signal beat into five components along each channel. The multi-channel multi-scale VCG tensor is formulated using the modes of each channel of VCG data, and it is used as the input to the deep convolutional neural network (CNN) to classify MI and normal sinus rhythm (NSR) classes. In the second stage, the multi-class deep CNN is used for the categorization of anterior MI (AMI), anterior-lateral MI (ALMI), anterior-septal MI (ASMI), inferior MI (IMI), inferior-lateral MI (ILMI), inferior-posterior-lateral (IPLMI) classes using MI detected multi-channel multi-scale VCG instances from the first stage. The proposed approach is developed using the VCG data obtained from a public database. The results reveal that the approach has obtained the accuracy, sensitivity, and specificity values of 99.58%, 99.18%, and 99.87%, respectively, for MI detection. Moreover, for MI localization, we have obtained the overall accuracy value of 99.86% in the second stage for our proposed network. The proposed approach has demonstrated superior classification performance compared to the existing VCG signal-based MI detection and localization techniques.

**Keywords:** myocardial infarction; vectorcardiogram; multivariate VMD; deep CNN; accuracy

## 1. Introduction

The obstruction in one of the coronary arteries of the heart causes the myocardial infarction (MI) disease [1,2]. Typically, the MI is progressed in three phases [3]. These three phases are (a) ischemic phase, (b) acute phase, and (c) myocardial necrosis phase. The 12-lead ECG signal is used in the clinical study for the early detection and localization of MI pathology [4]. The ST-segment elevations, inverted T-waves, and pathological Q-waves are the morphological changes observed in the ECG signals of different leads in MI pathology [5]. The morphological changes in the ECG signals of the channels or leads,

such as V1, V2, V3, and V4, are used to diagnose anterior MI (AMI) [6]. Similarly, inferior MI is diagnosed based on the variations in the morphologies of ECG signals for II, III, and aVF channels. Moreover, the morphological variations in the ECG signals of I, aVL, V5, and V6 channels are used to diagnose left lateral MI pathology [6]. In 12-lead ECG, no ECG lead capture the information about the diagnosis of posterior MI [6,7]. However, the reciprocal changes in the V1 and V2 channel ECG signals are used in the clinical study to diagnose posterior MI [8]. Vectorcardiogram (VCG) is an orthogonal three lead system which measures the heart's electrical activity along transverse, sagittal, and frontal planes, and it has been used for the detection of MI pathology [3,9]. The 12-lead ECG can be derived from the VCG signal using various transformation techniques [10]. In VCG, one of the orthogonal leads reveals the posterior view of the heart [11]. Hence, the method based on the analysis of VCG signal information is helpful to detect and localize MI pathology. The continuous recording and monitoring of VCG signal information for MI disease diagnosis is cumbersome, and hence automated approaches have been used for the accurate detection and localization of MI using VCG signals [3]. The development of novel approaches to detect and localize MI pathology using the VCG signals is challenging in clinical study.

In recent years, various approaches have been developed to detect MI using VCG signals [12–15]. The methods based on the evaluation of various VCG signal morphological features, such as difference in ST-T vector magnitude, area of ST-segment vector, and other T-wave features, have been used to detect MI disease [14,16–18]. Similarly, in [13], authors have applied independent component analysis (ICA) and principal component analysis (PCA) for projecting VCG signal feature vector into a lower-dimensional space. They have extracted various morphological features from the VCG signal to formulate the feature vector. The neural network-based classifier has been used for the detection of MI using reduced dimension feature vector of VCG signal [13]. In [14], authors have computed octant and vector-based features from VCG signals and used a decision tree model to detect MI pathology. These methods require the detection of P, Q, R, S, T-onset points manually in the VCG signal to compute the morphological features [3]. In literature, various wavelet-based techniques, such as multi-scale recurrent quantification analysis (MRQA) [15], and complex wavelet sub-band features [3] have been used to detect MI using VCG signals. In [15], each channel of the VCG signal is decomposed into sub-band signals using discrete wavelet transform (DWT). From each sub-band signal, the recurrent quantification analysis (RQA) based non-linear features have been extracted, and Gaussian discriminant analysis (GDA) classifier is used for the detection of MI [15]. Moreover, in [3], the dual-tree complex wavelet transform (DT-CWT) has been used to decompose the VCG signal into sub-band signals along each channel. The entropy and L1-norm features have been extracted from each sub-band signal. The relevance vector machine (RVM) classifier has been used to detect MI from these VCG signals features [3]. Along with cardiac signal processing, cardiac imaging today represents an important area of clinical research that has achieved excellent results in recent years, such as deep-learning approaches [19]; especially, this led to the development of computer-assisted tools capable of segmenting the whole heart [20,21], as well as identifying specific regions of interest [22]. In the wavelet-based approach, the pre-defined basis functions and the number of decomposition levels are used to compute sub-band signals from VCG signal [23]. Additionally, the mentioned VCG signal-based approaches have considered only for MI detection. The automated classification of various types of MI pathologies has not been considered using VCG signals. The existing VCG-based approaches have considered various feature extraction and machine learning methods to detect MI. In recent years, various deep learning-based approaches have been used to detect and localize MI using 12-lead ECG signals [2,24,25]. The deep learning-based MI detection and localization methods do not require extracting features from 12-lead ECG signals [2]. The deep learning-based methods have not been explored for the detection and localization of MI using VCG signals. Therefore, a deep learning-based approach can be developed to detect and localize MI using VCG signals.

The multivariate variational mode decomposition (MVMD) is a recently proposed signal processing technique to decompose the multi-channel signals into components or modes [26]. This method is fully signal-driven and does not consider any basis functions and decomposition levels like DWT to obtain components of non-stationary signals. The univariate version of VMD has been used for the analysis of ECG signals for the detection of ventricular tachycardia and atrial fibrillation episodes [27,28]. The VCG is a multi-channel signal, and, therefore, the MVMD can be used to decompose the signal into modes. Moreover, deep learning-based methods have been used in the multi-scale or modal domain of ECG signals to detect cardiac ailments [29]. For VCG signal, the deep learning method has not been explored in the multi-scale domain to detect and localization of MI. The novelty of this work is to develop a multi-channel multi-scale deep learning-based framework to detect and localize MI using VCG signals. The important contributions of this work are given as follows:

1. The MVMD is introduced to decompose the VCG signals into sub-band signals or modes;
2. A multi-channel multi-scale two-stage deep convolutional neural network (CNN) framework is proposed for the detection and localization of MI;
3. The MI types, such as AMI, IMI, ILMI, ALMI, ASMI, and IPLMI, are classified in the second stage of the proposed multi-channel multi-scale deep CNN (MMD-CNN) model;
4. The multi-channel multi-scale two-stage deep CNN performance is evaluated using hold-out and 10-fold cross-validation (CV) schemes.

The remaining sections of this paper are written as follows. The explanation regarding the VCG signal database is written in Section 2. In Section 3, the proposed approach for MI detection and localization is described. Section 4 presents the results and discussion of the proposed approach. In Section 5, conclusions of this paper is summarized.

## 2. VCG Signal Database

In this work, the VCG signals from the PTB diagnostic database (https://www.physionet.org/content/ptbdb/1.0.0/ (accessed on 20 June 2021)). were used to develop the proposed multi-channel multi-scale two-stage deep CNN approach [30,31]. The PTB database from Physionet comprises both 12-lead ECG and 3-lead VCG recordings of normal sinus rhythm (NSR) and various heart diseases, such as MI, hypertrophy, cardiomyopathy, bundle branch block, and dysrhythmia, respectively [30]. Each VCG signal has been sampled at 1000 samples per second in the PTB database, and the amplitude value of each lead VCG varied between $-16.384$ mV to $16.384$ mV. In this study, we have used 73 VCG recordings from 52 healthy controls (HC) subjects of PTB diagnostic database. Similarly, 99 VCG recordings from 148 subjects with MI pathology are used. For MI localization, 13, 20, 11, 21, 21, and 13 VCG recordings from AMI, IMI, ALMI, ASMI, ILMI, and IPLMI classes, respectively, are considered. In the PTB diagnostic database [30,31], the number of VCG recordings for MI class is higher than the healthy class. A higher difference in the number of VCG instances between MI and healthy classes may cause the over-fitting problem during the training of the proposed MMDCNN model. Due to this reason, we have considered only 99 VCG recordings from the MI class in this work. Each VCG recording in the PTB diagnostic database contains three orthogonal leads ($V_x$, $V_y$, $V_z$), which represent the electrical activity of heart in three different planes [28].

## 3. Method

The proposed MI detection and localization stages are shown in a flow-chart form in Figure 1a,b, respectively. The MI detection stage comprises the filtering of VCG signal, segmentation of VCG recordings into beats, decomposition of VCG beat into multi-scale VCG tensors using MVMD, and deep CNN to detect MI pathology. Similarly, the localization stage consists of the classification of AMI, IMI, ALMI, ASMI, ILMI, and IPLMI beats using

MI detected multi scale VCG tensor data. The following section briefly discuss each part of the flow-chart, as shown in Figure 1.

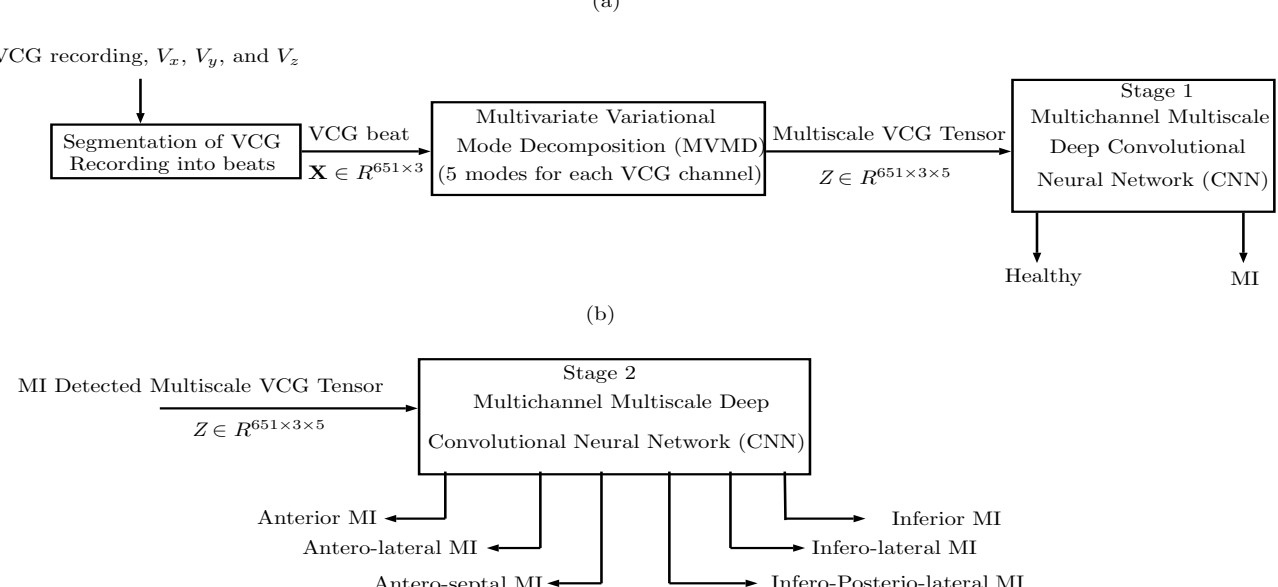

**Figure 1.** (**a**) Stage1 1: MI detection block using VCG signals. (**b**) Stage 2: MI localization using MI detected multi-scale VCG tensor.

### 3.1. Segmentation of VCG Data

In this work, we have performed the amplitude normalization for each lead of VCG recording [28]. The samples of raw VCG signal corresponding to each lead is divided by the maximum absolute value the signal to obtain the normalized VCG signal [6]. After normalization of each lead VCG recording, we have detected the R-peak in the $V_x$ lead of VCG signals. The beat by beat segmentation of each VCG recording is performed using a window of size 651 samples [2]. The 251 samples before each R-peak of the $V_x$ lead VCG signal and 400 samples after R-peak are considered for the beat by beat segmentation of VCG signal [2]. The number of MI and NSR VCG beats used for the proposed MI detection work are shown in Table 1. Similarly, the number of VCG beats evaluated for each type of MI are also shown in Table 1.

**Table 1.** Number of VCG beats used for MI detection and localization.

| Stage 1: MI detection | | | | | |
|---|---|---|---|---|---|
| Class | NSR | | | MI | |
| VCG beats | 9874 | | | 13982 | |
| **Stage 2: MI localization** | | | | | |
| Class | AMI | ALMI | ASMI | IMI | ILMI | IPLMI |
| VCG beats | 1664 | 1778 | 3057 | 2807 | 3049 | 1981 |

### 3.2. Multivariate VMD for VCG Signal Analysis

In this study, we have used MVMD to evaluate the modes of VCG beat along each orthogonal lead. The MVMD is the extension of VMD algorithm used to decompose multi-channel signals into modes [26]. The VCG beat is given as $v_m(n)$, with $n = 1, 2, ...N$. $N$ is the number of samples in the VCG beat. The parameter $m$ is denoted as the $m$th orthogonal lead of VCG beat. The VCG beat synthesized from its modes is given as follows:

$$v_m(n) = \sum_{i=1}^{k} u_m^i(n) \tag{1}$$

where, $u_m^i(n)$ is the $i$th mode of $m$th lead VCG signal $u_m^i(n) = \left[u_1^i(n), u_2^i(n), u_3^i(n)\right]$ is also interpreted as the multivariate modulated oscillations of VCG signal with $i = 1, 2, \ldots\ldots k$, and $k$ is the total number of modes [26]. The vector analytic representation of $i$th mode of $m$th lead VCG is written as follows [26]:

$$\tilde{u}_m^i(n) = u_m^i(n) + j\mathrm{H}\left(u_m^i(n)\right) \tag{2}$$

where, $\mathrm{H}\left(u_m^i(n)\right)$ is the Hilbert transform of $i$th mode of $m$th lead VCG signal [26]. In MVMD, the objective is to evaluate the modes of VCG signal based on the criteria as (a) the sum of bandwidth of components or modes of VCG should be minimum and (b) sum of all modes should recover the VCG signal along each lead [26]. The optimization problem of MVMD for the decomposition of VCG signal is formulated as follows [26]:

$$\min_{u_m^i(n), w^i} \quad \left\{ \sum_{i=1}^{k} \sum_{m=1}^{M} \left\| \frac{\partial}{\partial n} \left[ \tilde{u}_m^i(n) e^{-jw^i n} \right] \right\|_F^2 \right\}$$
$$\text{s.t.} \quad \sum_{i=1}^{k} u_m^i(n) = v_m(n), \quad m = 1, 2, \text{ and } 3 \tag{3}$$

where, $\|\bullet\|_F$ is the representation of Frobenious norm [26]. The optimization problem in Equation (3) can be reformulated using augmented Lagrangian and it is given as follows:

$$L\left\{u_m^i(n), w^i, \eta_m(n)\right\} = \beta \sum_{i=1}^{k} \sum_{m=1}^{M} \left\| \frac{\partial}{\partial n} \left[ u_m^i(n) e^{-jw^i n} \right] \right\|_F^2 + \sum_{m=1}^{M} \left\| v_m(n) - \sum_{i=1}^{k} u_m^i(n) \right\|_F^2 + \sum_{m=1}^{M} \left\langle \eta_m(n), v_m(n) - \sum_{i=1}^{k} u_m^i(n) \right\rangle \tag{4}$$

where, $\eta_m(n)$ is the Lagrangian multiplier for $m$th lead VCG beat, and $\beta$ is interpreted as the penalty factor for MVMD. The modes of VCG beat along each lead is iteratively evaluated based on the solution of Equation (4) using alternating direction method of multipliers (ADMM) [26]. The complete algorithm of MVMD for the extraction of modes from the non-stationary signals has been given in [26]. In this study, we have evaluated five modes from the VCG beat along each orthogonal lead. The multi-scale VCG tensor is formulated using the modes of VCG beat and the size of multi-scale VCG tensor is $651 \times 3 \times 5$.

For NSR class, the $V_x$, $V_y$, and $V_z$ lead VCG beat are shown in Figure 2a,g,m, respectively. The modes of $V_x$, $V_y$, and $V_z$ lead VCG beats evaluated using MVMD are shown Figure 2b–f,h–l,n–r, respectively. Similarly, the $V_x$, $V_y$, and $V_z$ channel VCG beats for IPLMI class are shown in Figure 3a,g,m, respectively. For IPLMI class, the modes of $V_x$, $V_y$, and $V_z$ lead VCG beats are depicted in Figure 3b–f,h–l,n–r, respectively. It can be observed from these plots that the modes of each lead VCG beat have different shape and amplitude values for IPLMI and NSR classes. In VCG signal, the clinical parameters, such as QRS-complex shape, special QRS-T angle, T-wave shape are different for healthy and MI cases [32]. The study in [33] has reported the physiological parameters of VCG signal for MI class, such as QRS-loop maximum vector magnitude, QRS-area perimeter ratio, and ST-vector magnitude, have higher mean values than those of healthy class. Similarly, the VCG parameters, such as QRS-loop volume, QRS-loop planar area, maximum of the distance between QRS-centroid and QRS-loop, and QRS-perimeter have the lowest mean values for MI class as compared to healthy class [33]. For the AMI case, there is abnormal posterior deviation in the QRS-vector of VCG signal [34]. Similarly, for the posterior-lateral MI case, the pathological changes, such as oriented T-loop and maximal leftward deviation of frontal plane QRS-vector are observed [35]. The transverse plane QRS-vector maximum value greater than 1.5 mV is also used as the criteria for the detection of inferior and posterior MI using VCG signals [11]. These differences in the morphological parameters of VCG signal for NSR and various types of MI cases can be captured in the modes which are evaluated using MVMD. Therefore, the deep CNN model designed using the modes of the VCG beat can be used to detect and localize MI.

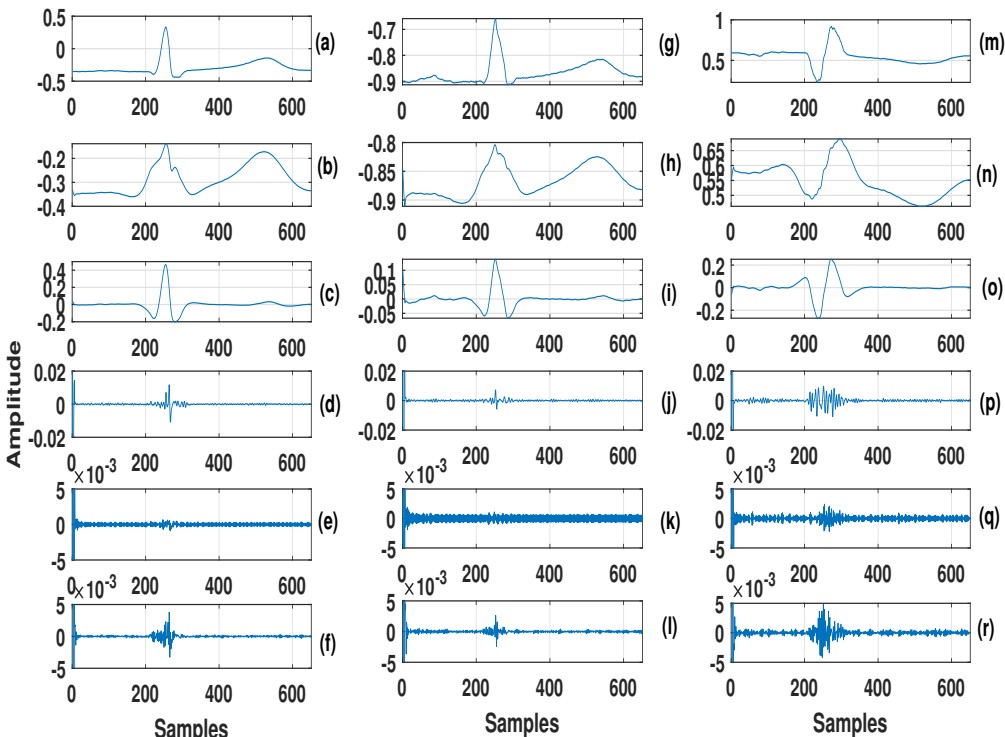

**Figure 2.** (**a**) $V_x$ lead VCG signal for NSR class. (**b–f**) mode 1 to 5 of $V_x$ lead VCG signal for NSR class. (**g**) $V_y$ lead VCG signal for NSR class. (**h–l**) mode 1 to 5 of $V_y$ lead VCG signal for NSR class. (**m**) $V_z$ lead VCG signal for NSR class. (**n–r**) mode 1 to 5 of $V_z$ lead VCG signal for NSR class.

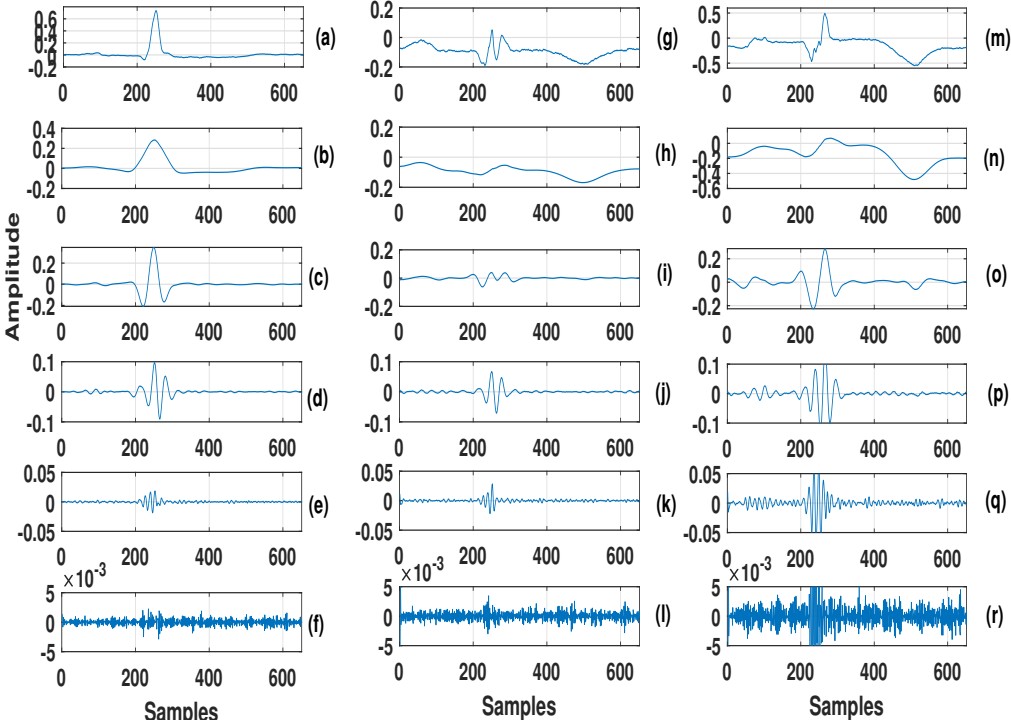

**Figure 3.** (**a**) $V_x$ lead VCG signal for IPLMI class. (**b–f**) mode 1 to 5 of $V_x$ lead VCG signal for IPLMI class. (**g**) $V_y$ lead VCG signal for IPLMI class. (**h–l**) mode 1 to 5 of $V_y$ lead VCG signal for IPLMI class. (**m**) $V_z$ lead VCG signal for IPLMI class. (**n–r**) mode 1 to 5 of $V_z$ lead VCG signal for IPLMI class.

### 3.3. Multi-Channel Multi-Scale Deep Convolutional Neural Network

In this work, a novel MMDCNN model is proposed to detect and localize MI. The Python codes for the MMDCNN model is available at (https://github.com/JayKarhade/MI_VCG_DL (accessed on 20 August 2021)). The MMDCNN architecture shown in Figure 4 comprises 12 layers. The first and last layers are interpreted as input and output layers of MMDCNN model. The input layer contains the multi-scale VCG tensor. The output layer consists of two neurons for MI detection stage, one for NSR class and the other for MI class. Similarly, for the MI localization stage, the output layer contains six neurons corresponding to six types of MI classes as AMI, IMI, ALMI, ASMI, ILMI, and IPLMI, respectively. The MMDCNN contains four convolutions, two max-pooling, and four dense layers for both MI detection and localization stages. The mathematical expression to compute the $t$th feature map for first convolution layer is given as follows [29,36]:

$$\mathbf{X}_t^{(l)}(\widetilde{n}) = h\left(\sum_{n=1}^{N}\sum_{m=1}^{M}\sum_{i=1}^{I}\mathbf{X}(n,m,i)\mathbf{K}_t\left(\widetilde{n}-n+\frac{N}{2},m,i\right)+b_t\right) \tag{5}$$

where $\mathbf{X}(n,m,i)$ is the input to the MMDCNN and $i = 1,2\ldots\ldots I$ and $m = 1,2\ldots M$, respectively. The parameters $I$ and $M$ are total number of modes and channels, respectively. Similarly, the mathematical expression for the evaluation of feature maps in other convolution layers are evaluated as follows [29,36]:

$$\mathbf{X}_t^{(l)}(\widetilde{n}) = h\left(\sum_{n=1}^{N}\sum_{c=1}^{C}\mathbf{X}_{\widetilde{t}}^{(l-1)}(n,c)\widetilde{\mathbf{K}}_t\left(\widetilde{n}-n+\frac{N}{2},c\right)+\widetilde{b}_t\right) \tag{6}$$

$\mathbf{X}_{\widetilde{t}}^{(l-1)}(n,c)$ is the $\widetilde{t}$th feature map at $(l-1)$th convolution layer. Similarly, the feature maps for second, third and fourth convolution layers are evaluated using Equation (6). The $\mathbf{X}_t^{(l)}$ is denoted as the $t$th feature map for $l$th convolution layer. Moreover, the mathematical expression to evaluate the pooling layer feature map is given as follows [29,36]:

$$\mathbf{X}_t^{(l)}(\widetilde{n}) = \text{max-pooling}(\mathbf{X}_t^{(l-1)}(\widetilde{n})) \tag{7}$$

For dense layers, the feature vector is evaluated as follows [37]:

$$\mathbf{a}^{(l)} = h(\mathbf{a}^{(l-1)}\overline{\mathbf{W}}^{(l)} + \overline{b}^{(l)}) \tag{8}$$

where $\mathbf{a}^{(l)}$ is the feature vector for $l$th dense layer. $\overline{\mathbf{W}}^{(l)}$ is the weight matrix between $(l-1)$th dense and $l$th dense layers [37]. $\overline{b}^{(l)}$ is the bias for $l$th dense layer. The categorical cross-entropy-based cost function is used for MMDCNN for both detection and localization stages [38]. The hyper-parameters used for MMDCNN in detection and localization stages are shown in Table 2. In this study, for both MI detection and localization stages, the hold-out validation and 10-fold cross-validation (CV) methods [37] are used to select the training and test VCG beats. For hold-out validation 78.75%, 11.25%, and 10% VCG beats are used as training, validation, and testing, respectively, for MMDCNN model during detection and localization phases. We have used the performance measures such as accuracy, sensitivity, specificity, and Kappa scores for the MI detection using MMDCNN classifier [37,39]. Similarly, for MI localization, the overall accuracy (OA), individual accuracy (IA), and Kappa score are used to evaluate the performance in the second stage MMDCNN [6].

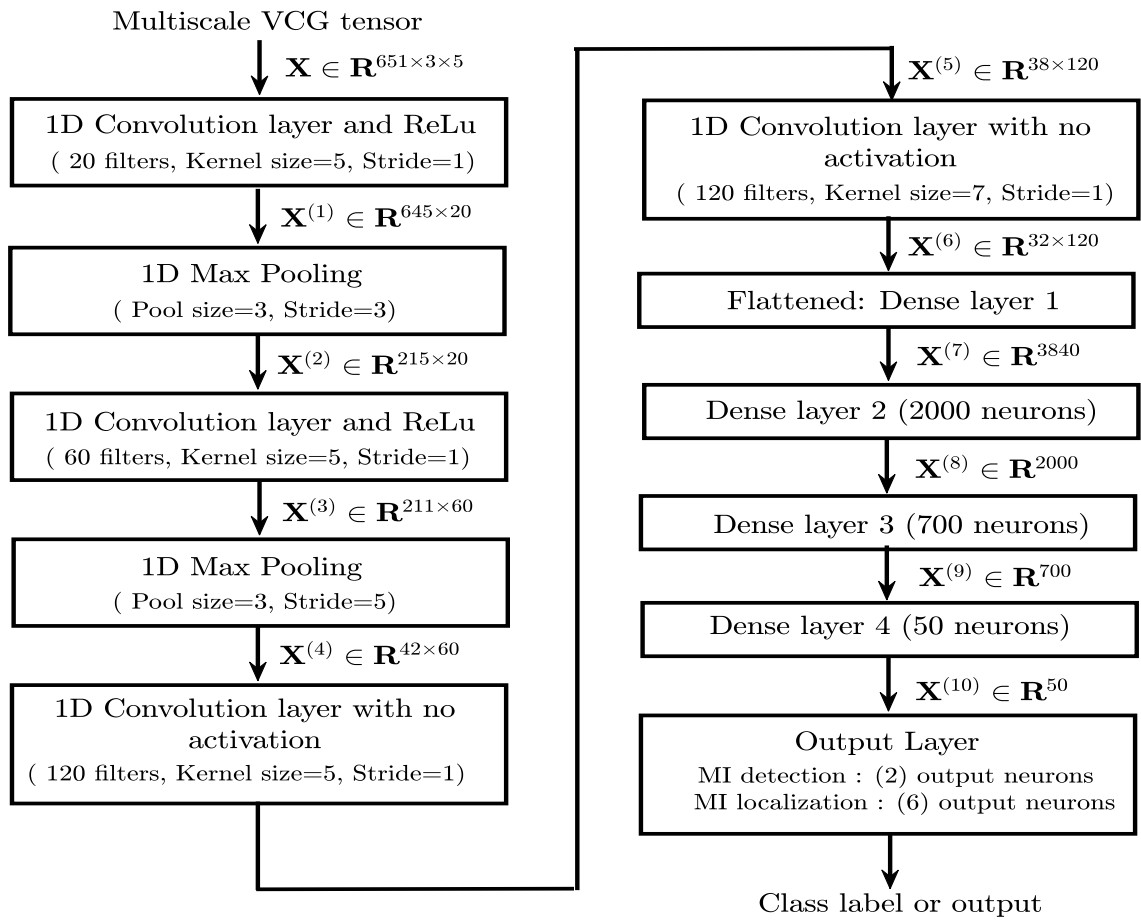

**Figure 4.** Proposed MMDCNN model to detect and localize MI using VCG beats.

**Table 2.** Hyper-parameters used MI detection and localization using our proposed MMDCNN model.

| | Hold-out (MI Detection) | | | | 10-fold (MI Detection ) | | |
|---|---|---|---|---|---|---|---|
| Parameters | Optimizer | Batch size | Epochs | Learning rate | Batch size | Epochs | Learning rate |
| Values | Adam | 1024 | 15 | 0.0001 | 1024 | 15 | 0.0001 |
| | Hold-out (MI Localization) | | | | 10-fold (MI Localization ) | | |
| Parameters | Optimizer | Batch size | Epochs | Learning rate | Batch size | Epochs | Learning rate |
| Values | Adam | 1024 | 15 | 0.00004 | 256 | 15 | 0.0001 |

## 4. Results and Discussions

The results evaluated using the proposed MMDCNN for MI detection and localization using VCG signals are shown in this section. In Table 3, we have shown the accuracy, sensitivity, specificity, and kappa score values for our proposed MMDCNN model with hold-out CV. Similarly, for MI detection, the accuracy vs. epoch plots for training and validation VCG instances obtained using MMDCNN are illustrated in Figure 5. It is evident from this plot that both training and validation accuracy values are 100% after 10th epoch. Similarly, we have shown the confusion matrix obtained using the proposed MMDCNN for MI detection using VCG signals for one random hold-out trial in Table 4. The number of false-positive and false-negative values are 1 in the confusion matrix table. The accuracy, sensitivity, specificity, and kappa values for this random hold-out validation are 99.9%, 99.89%, 99.92%, and 0.998, respectively. The average values of accuracy, sensitivity, specificity, and kappa scores over five trial-based random validation are more than 99% (as seen from Table 3).

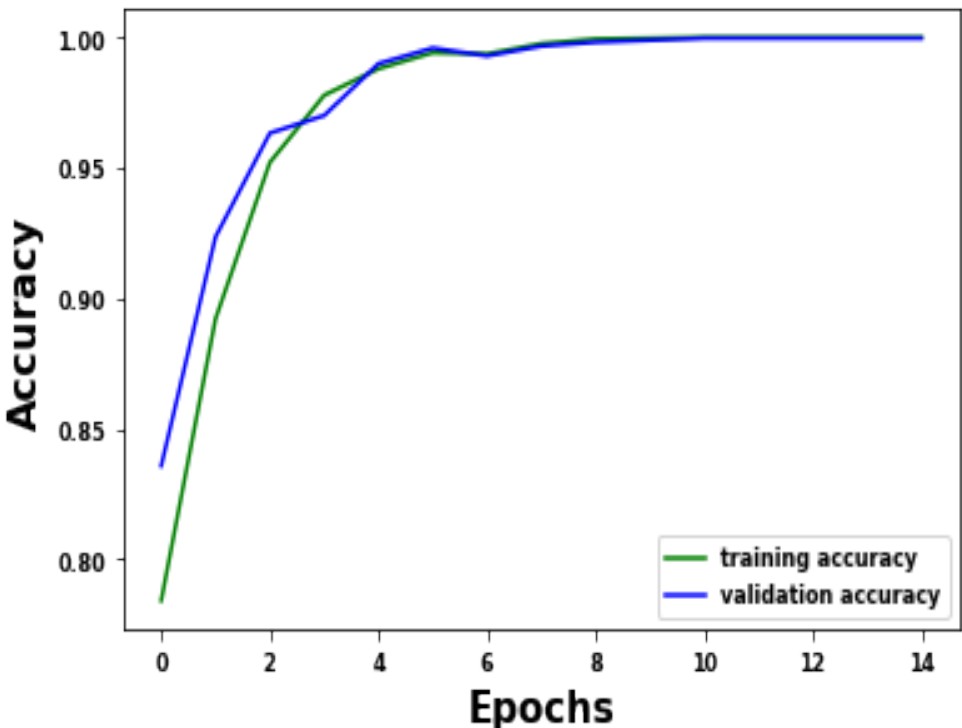

**Figure 5.** Graphs of accuracy vs. epochs of multi-channel multi-scale deep CNN obtained using training and validation VCG instances for MI detection.

**Table 3.** Classification results of two class multi-channel multi-scale deep CNN obtained for the detection of MI using hold-out validation.

| Accuracy (%) | Sensitivity (%) | Specificity (%) | Kappa |
|:---:|:---:|:---:|:---:|
| 99.58 ± 0.38 | 99.18 ± 0.90 | 99.87 ± 0.07 | 0.990 ± 0.01 |

**Table 4.** Confusion matrix for one-trail of hold-out CV for MI detection.

| | | Predicted | |
|:---:|:---:|:---:|:---:|
| | | **Healthy** | **MI** |
| Actual | Healthy | 978 | 1 |
| | MI | 1 | 1390 |

For MI detection, the classification results obtained for the proposed first stage MMDCNN using 10-fold CV are shown in Table 5. It can be observed from this table that, the accuracy values are more than 99.50% for each fold. Similar high percentages in the sensitivity and specificity are seen in each fold using the first stage MMDCNN method for MI detection. It can also be observed that the Cohen kappa score is more than 0.99 for each fold. From these 10-fold CV results, It can be noted that the proposed first stage deep CNN successfully detected MI using the modes of VCG beats.

**Table 5.** Results obtained using multi-channel multi-scale deep CNN with 10-fold CV.

| Folds | 1 | 2 | 3 | 4 | 5 | 6 | 7 | 8 | 9 | 10 | Value ($\mu \pm \sigma$) |
|:---:|:---:|:---:|:---:|:---:|:---:|:---:|:---:|:---:|:---:|:---:|:---:|
| Accuracy (%) | 99.7 | 100 | 99.95 | 100 | 99.74 | 99.95 | 100 | 100 | 100 | 100 | 99.93 ± 0.11 |
| Sensitivity (%) | 100 | 100 | 100 | 100 | 99.38 | 100 | 100 | 100 | 100 | 100 | 99.93 ± 0.19 |
| Specificity (%) | 99.49 | 100 | 99.99 | 100 | 100 | 99.92 | 100 | 100 | 100 | 100 | 99.94 ± 0.16 |
| Kappa | 0.993 | 1 | 0.999 | 1 | 0.994 | 0.999 | 1 | 1 | 1 | 1 | 0.998 ± 0.002 |

The confusion matrix obtained using one random trial-based hold-out validation for MI localization with second stage MMDCNN is shown in Table 6. Similarly, we have shown the accuracy vs. epoch plots for training and validation of multi-scale VCG tensor instances in Figure 6. It can be observed from these plots that both training and validation accuracy values obtained are more than 99% after 10th epoch using the second stage MMDCNN model. It can be seen from Table 6 that the number of true positives for AMI, IMI, ALMI, ASMI, ILMI, and IPLMI classes are obtained as 162, 284, 185, 287, 301, and 201, respectively. Three multi-scale VCG tensor instances, which belong to IMI, are classified as ALMI class. Similarly, the classification results of the proposed second-stage MMDCNN obtained for MI localization using hold-out validation are shown in Table 7. It can be noted that the average IA values are more than 99% for AMI, IMLI, ALMI, ILMI, and IPLMI classes. For ASMI class, the IA value is 94.38%. The OA and kappa values obtained are 98.77% and 0.982, respectively, using the proposed second-stage MMDCNN model.

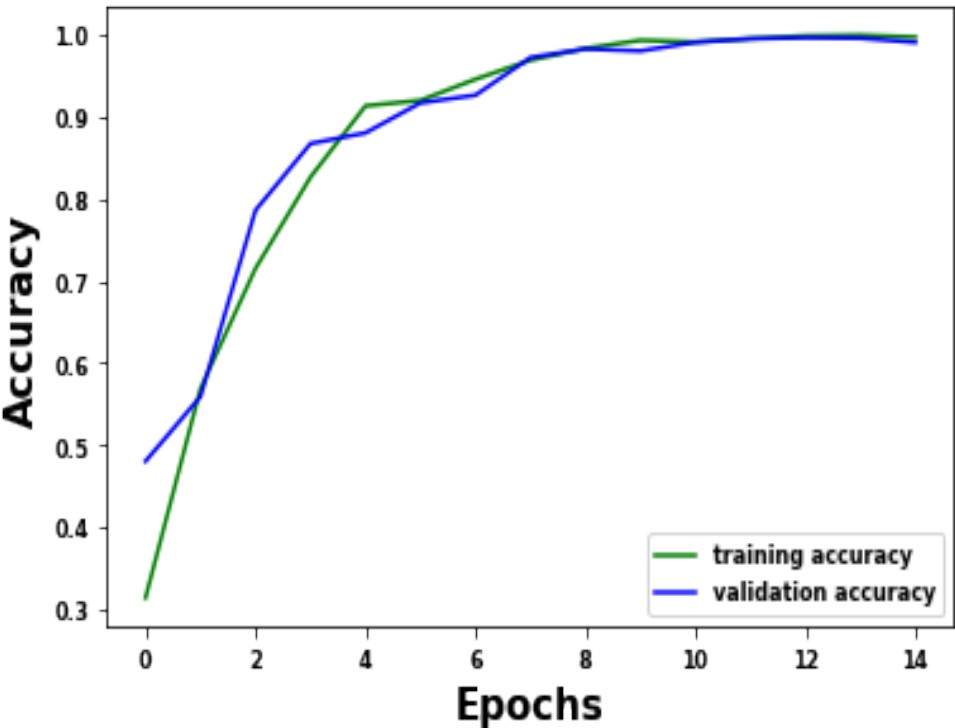

**Figure 6.** Plots of accuracy vs. epochs of multi-channel multi-scale deep CNN obtained using training and validation VCG instances for MI localization.

**Table 6.** Confusion matrix obtained using the proposed second stage multi-channel multi-scale deep CNN classifier for MI localization.

| | | Predicted Classes | | | | | |
|---|---|---|---|---|---|---|---|
| | | **AMI** | **ALMI** | **ASMI** | **IMI** | **ILMI** | **IPLMI** |
| | AMI | 162 | 0 | 0 | 0 | 0 | 0 |
| | ALMI | 0 | 185 | 0 | 0 | 1 | 0 |
| Actual Classes | ASMI | 0 | 0 | 287 | 0 | 0 | 0 |
| | IMI | 0 | 3 | 0 | 284 | 0 | 0 |
| | ILMI | 0 | 0 | 0 | 0 | 301 | 1 |
| | IPLMI | 0 | 0 | 0 | 0 | 0 | 200 |

**Table 7.** Classification results obtained for MI localization using proposed MMDCNN model with hold-out validation.

| Parameters | Value ($\mu \pm \sigma$) |
|---|---|
| IA$_{AMI}$ (%) | $99.79 \pm 0.35$ |
| IA$_{ALMI}$ (%) | $99.64 \pm 0.31$ |
| IA$_{ASMI}$ (%) | $94.38 \pm 5.56$ |
| IA$_{IMI}$ (%) | $99.53 \pm 0.53$ |
| IA$_{ILMI}$ (%) | $99.66 \pm 0.00$ |
| IA$_{IPLMI}$ (%) | $99.66 \pm 0.57$ |
| OA (%) | $98.77 \pm 0.96$ |
| Kappa | $0.982 \pm 0.014$ |

Moreover, we have shown the classification results of second-stage MMDCNN for MI localization using a 10-fold CV and these results are shown in Table 8. It can be observed from these results that for ILMI class, the accuracy value of each fold is more than 99%. Similarly, for IMI class, apart from 5th fold, more than 99% accuracy values are observed for other folds. For IPLMI and AMI classes, more than 98% accuracy values are obtained in each fold using second stage MMDCNN model. Similarly, more than 97% accuracy values are obtained using MMDCNN classifier for ASMI and and ALMI classes. The overall accuracy (OA) values are obtained as more than 99% at each fold. The kappa value of more than 0.97 is observed for each fold using MMDCNN classifier.

**Table 8.** Classification results obtained for MI localization using proposed MMDCNN model with 10-fold CV.

| Folds | 1 | 2 | 3 | 4 | 5 | 6 | 7 | 8 | 9 | 10 | $\mu \pm \sigma$ |
|---|---|---|---|---|---|---|---|---|---|---|---|
| IA$_{AMI}$ (%) | 100 | 98.79 | 100 | 99.4 | 100 | 100 | 100 | 100 | 100 | 100 | $99.81 \pm 0.40$ |
| IA$_{ALMI}$ (%) | 99.43 | 97.75 | 97.75 | 96.57 | 99.43 | 98.31 | 97.19 | 99.43 | 97.75 | 98.3 | $98.19 \pm 0.99$ |
| IA$_{ASMI}$ (%) | 97.38 | 95.09 | 100 | 97.37 | 100 | 100 | 99.67 | 98.69 | 99.01 | 100 | $98.72 \pm 1.64$ |
| IA$_{IMI}$ (%) | 100 | 99.64 | 100 | 99.28 | 98.93 | 99.64 | 98.57 | 100 | 99.64 | 99.28 | $99.49 \pm 0.48$ |
| IA$_{ILMI}$ (%) | 99.67 | 99.66 | 99.34 | 99.67 | 99.01 | 99.67 | 99.01 | 99.67 | 100 | 100 | $99.57 \pm 0.34$ |
| IA$_{IPLMI}$ (%) | 99.49 | 100 | 98.98 | 98.48 | 99.49 | 99.49 | 98.48 | 100 | 98.98 | 98.48 | $99.18 \pm 0.59$ |
| OA(%) | 99.72 | 99.09 | 99.44 | 99.23 | 99.37 | 99.65 | 98.88 | 99.58 | 99.37 | 99.44 | $99.37 \pm 0.25$ |
| Kappa | 0.990 | 0.978 | 0.993 | 0.979 | 0.993 | 0.994 | 0.986 | 0.994 | 0.991 | 0.993 | $0.989 \pm 0.006$ |

The classification results of MMDCNN models evaluated using the selected modes of each lead VCG signal, and all modes of high-pass filtered VCG signals for MI detection with hold-out validation are shown in Table 9. It is observed that the average accuracy value of MMDCNN is 99.58% using mode 1 and mode 2 of each lead VCG signal. The average accuracy value remains the same as the accuracy of MMDCNN model using all modes of VCG signals for MI detection. Mode 1 and mode 2 capture the significant information of the VCG signal after decomposition using MVMD. Henceforth, the accuracy value remains the same for MI detection using selected modes and the MMDCNN classifier. Moreover, we have also evaluated the classification performance of the MMDCNN model using all modes of high-pass filtered VCG signal for MI detection. A high-pass Butterworth filter with a cut-off frequency of 0.5 Hz is applied to each lead VCG signal to remove baseline wondering artifacts [6,28]. It is observed from Table 9 that average accuracy, average kappa score, average sensitivity, and average specificity values are improved after the filtering of baseline wandering artifact from VCG signals. In Table 10, we have shown the individual accuracy value for each MI class, OA, and kappa scores of MMDCNN classifier for MI localization using mode 1 and mode 2 of each lead VCG signal and all modes of high-pass filtered VCG signals, respectively. It is observed that the OA value obtained using the MMDCNN model is less using mode 1 and mode 2 of VCG signals as compared to all modes of VCG signals. Similarly, the OA and kappa values are improved using the modes of high-pass filtered VCG signals with the MMDCNN classifier. For MI localization, the IA values for ASMI, IMI, and ILMI classes are also improved using the

modes of high-pass filtered VCG signals composed with the MMDCNN classifier. Moreover, we have also evaluated the classification results of the MMDCNN classifier using all modes of VCG signal with leave one out (LOO) CV strategy. The VCG beats of one recording are considered during testing of the MMDCNN model, whereas the VCG beats of all other VCG recordings are used to train the MMDCNN classifier. The same procedure is applied to all VCG recordings, and it can also be interpreted as a 171-fold CV strategy. The LOO CV or pre-recording-based MI detection results are shown in Figure 7. It is observed that out of 172 VCG recordings, 114 recordings are correctly classified with 100% accuracy. The OA value obtained using MMDCNN classifier with LOO CV strategy is 87.65%.

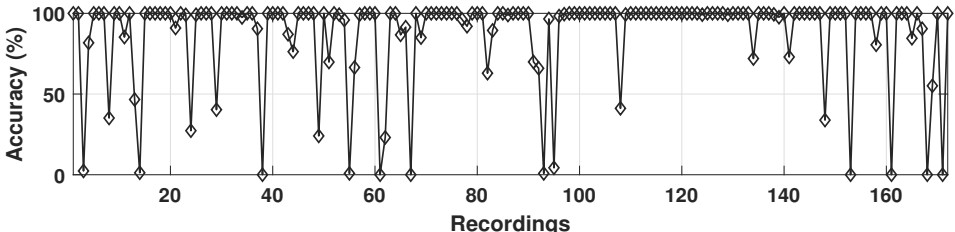

**Figure 7.** Accuracy values obtained using the proposed MMDCNN classifier for MI detection with leave one out CV strategy.

**Table 9.** Classification results of MMDCNN obtained using selected modes and all modes for the detection of MI.

| Mode Selection | Accuracy (%) | Sensitivity (%) | Specificity (%) | Kappa |
|---|---|---|---|---|
| Mode 1 and mode 2 of each lead VCG signals | 99.58 ± 0.37 | 99.17 ± 0.89 | 99.86 ± 0.06 | 0.991 ± 0.007 |
| All modes from high-pass filtered VCG signals | 99.92 ± 0.02 | 99.84 ± 0.05 | 100 ± 0 | 0.998 ± 0.0005 |

**Table 10.** Classification results obtained for MI localization using proposed MMDCNN model with mode selection.

| Parameters | Mode 1 and Mode 2 from Each Lead VCG Signals | All Modes from High-Pass Filtered VCG Signals |
|---|---|---|
| $IA_{AMI}$ (%) | 100 ± 0 | 100 ± 0 |
| $IA_{ALMI}$ (%) | 99.17 ± 0.29 | 98.55 ± 2.22 |
| $IA_{ASMI}$ (%) | 93.09 ± 5.16 | 99.26 ± 0.27 |
| $IA_{IMI}$ (%) | 99.30 ± 0.28 | 100 ± 0 |
| $IA_{ILMI}$ (%) | 99.50 ± 0.19 | 99.86 ± 0.14 |
| $IA_{IPLMI}$ (%) | 99.25 ± 0.50 | 98.45 ± 2.45 |
| OA (%) | 98.37 ± 1.43 | 99.44 ± 0.56 |
| Kappa | 0.976 ± 0.014 | 0.992 ± 0.006 |

We have also formulated the seven-class classification scheme as (Healthy vs. AMI vs. ALMI vs. ASMI vs. IMI vs. ILMI vs. IPLMI) using MMDCNN classifier with all modes of VCG signals. The seven class classification results obtained using the MMDCNN model are shown in Table 11. It can be observed that for healthy, ALMI, IMI, and IPLMI classes, the IA values are 69.87%, 83.22%, 82.61%, and 41.03%, respectively. The OA value of MMDCNN classifier obtained is 81.48%, which is less than the proposed two-stage MMDCNN model for MI detection and localization.

**Table 11.** Classification results obtained using MMDCNN classifier for seven class classification scheme with hold-out validation.

| Parameters | Value ($\mu \pm \sigma$) |
|:---:|:---:|
| Healthy (%) | $69.87 \pm 28.87$ |
| IA$_{AMI}$ (%) | $99.46 \pm 0.38$ |
| IA$_{ALMI}$ (%) | $83.22 \pm 10.14$ |
| IA$_{ASMI}$ (%) | $99.63 \pm 0.25$ |
| IA$_{IMI}$ (%) | $82.61 \pm 7.93$ |
| IA$_{ILMI}$ (%) | $98.44 \pm 2.28$ |
| IA$_{IPLMI}$ (%) | $41.03 \pm 18.18$ |
| OA (%) | $81.48 \pm 0.63$ |
| Kappa | $0.777 \pm 0.008$ |

The classification performance of the proposed first stage MMDCNN classifier is compared with the existing techniques for MI detection using VCG signals with a 10-fold CV-based technique. The comparison results are shown in Table 12. The work reported in [15] has computed features from each lead of VCG signal using multi-scale recurrent quantification analysis (MRQA). The Gaussian discriminant analysis (GDA) based classification model has been used to detect MI using MRQA based VCG features. The sensitivity and specificity values of 96.50% and 75% have been obtained in their work. Similarly, in [14], the combination of octant and vector-based features have been obtained using VCG signal. The classification and regression tree (CART) based model has been used for the detection of MI. The classification performance, such as the sensitivity and specificity values of 97.28% and 96%, respectively, are reported. The complex wavelet sub-band features of VCG coupled with the RVM classifier have obtained the sensitivity and specificity values of 98.40%, and 98.66%, respectively, for MI detection [28]. The proposed MMDCNN model has obtained better classification performance than the existing machine learning-based methods for MI detection using VCG signals. The advantages of our proposed MMDCNN based approach are given as follows:

- A novel two-stage based MMDCNN model is proposed to detect and localize MI using VCG beats;
- The multi-scale analysis of VCG signal is performed using MVMD based multi-variate signal driven approach;
- The approach has demonstrated more than 99% accuracy for MI detection;
- The extraction of raw features from VCG signals are not required using the proposed approach for both detection and localization stages;
- The second stage MMDCNN model successfully classified six types of MI with an accuracy of more than 99%.

**Table 12.** Comparison of proposed MI detection approach with existing methods obtained using VCG signals (with 10-fold CV).

| Authors | Features Extracted | Classifiers Used | Sensitivity (%) | Specificity (%) |
|:---:|:---:|:---:|:---:|:---:|
| Yang et al., 2012 [14] | Octant and vector features evaluated from VCG signal | CART | 97.28 | 95 |
| Yang, 2011 [15] | DWT domain RQA features from VCG Signal | GDA | 96.50 | 75 |
| Tripathy et al., 2017 [28] | Complex wavelet sub-band features from VCG signal | RVM | 98.40 | 98.66 |
| Proposed work | Multi-channel and multi-scale domain learnable features | CNN | 99.93 | 99.94 |

In this work, the proposed approach has considered only 99 VCG recordings from different MI classes in the second stage for MI localization. The approach can be tested using VCG recordings from a huge database containing more subjects. The MVMD based multi-scale approach is used in this study to decompose the VCG signal. The other multi-scale analysis methods, such as multivariate empirical mode decomposition (MEMD) [40], multivariate projection based empirical wavelet transform (MPEWT) [41], and fast and adaptive based MEMD [42] can be used for the decomposition of VCG signals.

## 5. Conclusions

The multi-channel multi-scale two-stage deep CNN model is proposed to detect and localize MI using VCG signals. The MVMD is used to decompose the VCG beat into modes along with each orthogonal lead. The multi-channel multi-scale VCG tensor has been formulated and used as input to the deep CNN model to detect and localize MI. For MI detection, the proposed first-stage MMDCNN model obtained an average accuracy value of 99.93% with 10-fold CV. The second-stage MMDCNN model produced an average overall accuracy (OA) value of 99.37% for MI localization. The average OA values are more than 99% for AMI, IMI, ILMI, and ILMI classes. The proposed first-stage MMDCNN classifier obtained a higher accuracy value than the existing VCG based approaches for MI detection. The MMDCNN model can also be explored to detect other cardiac ailments, such as atrial fibrillation, hypertrophy, cardiomyopathy, ventricular arrhythmia, and bundle branch block using VCG signals.

**Author Contributions:** Conceptualization, J.K.; Data curation, R.K.T. and S.K.G.; Formal analysis, J.K. and U.R.A.; Methodology, P.G. and R.K.T.; Project administration, U.R.A.; Resources, R.K.T.; Supervision, R.K.T.; Writing—original draft, R.K.T., P.G.; Writing—review and editing, U.R.A. All authors have read and agreed to the published version of the manuscript.

**Funding:** This research received no external funding.

**Institutional Review Board Statement:** Not applicable.

**Informed Consent Statement:** Not applicable.

**Data Availability Statement:** The codes of the work is available at (https://github.com/JayKarhade/MI_VCG_DL (accessed on 20 August 2021)).

**Conflicts of Interest:** The authors declare no conflict of interest.

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
