# Peer review of "Multichannel Multiscale Two-Stage Convolutional Neural Network for the Detection and Localization of Myocardial Infarction Using Vectorcardiogram Signal"

_applsci, doi:10.3390/app11177920_

Round 1
Reviewer 1 Report
This study shows the potential of high-dimensional feature extraction for VCG or even ECG signals using method-based approach e.g., multivariate variational mode decomposition. It seems to successfully regularize CNN parameters to avoid overfitting and achieve better performance for MI detection and localization. However, there are critical issues that need to be confirmed. First, in PTB diagnostic database, each patient may have multiple ECG recordings, and thus, there must be a total of more than 148 ECG recordings for MI patients. Accordingly, this study seems to exclude some data and what are the decision criteria? Second, T loops are the dominant components in VCG signals to diagnose myocardial infarction, and the main frequency band of T loops will reach to 0.05 Hz. Therefore, the cut-off frequency of high pass filter 0.5 Hz may damage the information of myocardial repolarization. Third, the third to fifth modes of VCG signals extracted from MVMD are high frequency components which center frequency is much higher than T loops’. I’m wondering how these modes contribute themselves to CNN to classify correctly. Perhaps demonstrating the gradient of weights or excluding these modes and observing the fluctuation in accuracy can explain. Lastly, this study seems to directly segment VCG signals from all patients into training dataset and testing dataset for CNN input. This will cause data from one patient exist in both training and testing dataset and lead to data contamination called intra-patient problem. For this situation, the accuracy of CNN cannot faithfully present its universality and usually fail to classify correctly using different database. Therefore, the aothors must make sure the VCG data from one patient only exist in training or testing dataset, not both. Under the inter-patient scheme, the performance of CNN model is finally real at this time.

Author Response
Replies to the comments of the reviewer
Reviewer #1: This study shows the potential of high-dimensional feature extraction for VCG or even ECG signals using a method-based approach e.g., multivariate variational mode decomposition. It seems to successfully regularize CNN parameters to avoid overfitting and achieve better performance for MI detection and localization. However, there are critical issues that need to be confirmed.
1. First, in the PTB diagnostic database, each patient may have multiple ECG recordings, and thus, there must be a total of more than 148 ECG recordings for MI patients. Accordingly, this study seems to exclude some data, and what are the decision criteria?
Reply: We thank the reviewer for this comment. In this study, we have considered 99 VCG recordings (13 AMI, 20 IMI, 11 ALMI, 21 ASMI, 21 ILMI, and 13 IPLMI VCG recordings) from the MI class. In the PTB diagnostic database, the number of VCG recordings for the MI class is higher than the healthy class. A higher difference in the number of VCG beats between MI and healthy classes may cause the overfitting problem during the training of the proposed MMDCNN model. Due to this reason, we have considered only 99 VCG recordings from the MI class in this work. The corrections are incorporated on page number 3 and highlighted in magenta color.
2.Second, T loops are the dominant components in VCG signals to diagnose myocardial infarction, and the main frequency band of T loops will reach 0.05 Hz. Therefore, the cut-off frequency of the high pass filter 0.5 Hz may damage the information of myocardial repolarization.
Reply: We thank the reviewer for this comment. We have checked the code again. The high-pass filter has not been used in this work for the filtering of each channel VCG signal. The pre-processing step such as amplitude normalization is performed for each channel VCG signal.
However, we have also evaluated the classification performance of the MMDCNN model using the modes of high-pass filtered VCG signals. The results are shown in Table 9 and Table 10 for MI detection and localization. The corrections are incorporated on page number 12 and highlighted in magenta color.
3. Third, the third to fifth modes of VCG signals extracted from MVMD are high-frequency components whose center frequency is much higher than T loops’. I’m wondering how these modes contribute themselves to CNN to classify correctly. Perhaps demonstrating the gradient of weights or excluding these modes and observing the fluctuation in accuracy can explain.
Reply: This is a very good comment. As suggested by the reviewer, we have evaluated the classification performance of the MMDCNN classifier using mode 1 and mode 2 of each lead VCG signal for both MI detection and localization. The results are added in Table 9 and Table 10, respectively. The corrections are incorporated on page number 12 and highlighted in magenta color.
4. Lastly, this study seems to directly segment VCG signals from all patients into the training dataset and testing dataset for CNN input. This will cause data from one patient to exist in both training and testing datasets and lead to data contamination called intra-patient problems. For this situation, the accuracy of CNN cannot faithfully present its universality and usually fail to classify correctly using different databases. Therefore, the authors must make sure the VCG data from one patient only exist in the training or testing dataset, not both. Under the inter-patient scheme, the performance of the CNN model is finally real at this time.
Reply: We thank the reviewer for this comment. As suggested by the reviewer, we have added the leave-one-out CV results in the revised manuscript. The accuracy value for each recording is evaluated and it is shown in Fig. 7. The description of Fig.7 is highlighted in magenta color in the revised manuscript.
Reviewer 2 Report
This manuscript deals with detecting and categorizing Myocardial Infarction from vector cardiogram data by using deep convolutional neural network. It is written with scientific rigor, and the results look very promising. It is recommended for publication, once the questions below are successfully answered and minor errors are corrected.
Q1. Authors have used two stage strategy, where first stage distinguishes MI population from NSR, and second stage uses only MI group data to distinguish among six different subcategories. How does it compare with a single stage strategy, where NSR plus six MI subcategories constitute overall seven categories? How would it affect MI detection accuracy?
Q2. Authors have treated six MI subcategories as just different names, without describing any physiology or graphical traits each category has. It naturally leads to poor discussion regarding the non-uniformity of accuracy values. In authors’ view, why is the IA value of ASMI significantly lower than that of other subcategories? Any inputs from collaborating medical doctors?
<Minor issues>
- Abstract : Acronyms for different categories of MI should be corrected. (ALMI, IMI, etc)
- The position of Table captions need to be consistent. For example, does Table 4 caption refer to the table above it or below?
- Line 99 : “… proposed for detection and localization of MI.”
- Line 116 : Take care of the numbers. There are seven numbers listed, when there are only six different categories.
- Line 171 : Take care of Table number
- Line 178 : It can also be observed …
Author Response
Reviewer #2: This manuscript deals with detecting and categorizing Myocardial Infarction from vector cardiogram data by using deep convolutional neural networks. It is written with scientific rigor, and the results look very promising. It is recommended for publication, once the questions below are successfully answered and minor errors are corrected.
Reply: We thank the reviewer for the recommendation of our manuscript. As suggested by the reviewer, we have added corrections to the revised manuscript. In the revised manuscript, the replies to the comments of this reviewer are highlighted in blue color.
- Authors have used a two-stage strategy, where the first stage distinguishes the MI population from NSR, and the second stage uses only MI group data to distinguish among six different subcategories. How does it compare with a single-stage strategy, where NSR plus six MI subcategories constitute overall seven categories? How would it affect MI detection accuracy?
Reply: We thank the reviewer for this comment. As suggested by the reviewer, we have added the classification results of the MMDCNN classifier for seven class classification tasks in the revised manuscript. A new table (Table 11) has been added to the revised manuscript. The descriptions of Table 11 are highlighted in blue color on page 13 of the revised manuscript.
- Authors have treated six MI subcategories as just different names, without describing any physiology or graphical traits each category has. It naturally leads to poor discussion regarding the non-uniformity of accuracy values. In the authors' view, why is the IA value of ASMI significantly lower than that of other subcategories? Any inputs from collaborating medical doctors?
Reply: We thank the reviewer for this comment. As suggested by the reviewer, we have added the physiological descriptions regarding the variations in the various morphological parameters of VCG signals during different types of MI. We have collaborated with few cardiologists in Hyderabad, India, for ECG and VCG analysis during different types of MI. In the future, new deep learning approaches can be developed to detect and classify various cardiac diseases using VCG and ECG signals.
<Minor issues>
- Abstract: Acronyms for different categories of MI should be corrected. (ALMI, IMI, etc)
Reply: As suggested by the reviewer, we have corrected this mistake in the abstract section.
- The position of Table captions needs to be consistent. For example, does Table 4 caption refer to the table above it or below?
Reply: We have added the Table 4 caption in the revised manuscript. The corrections are highlighted in blue color in the revised manuscript.
- Line 99: “… proposed for detection and localization of MI.”
Reply: We have corrected this grammatical mistake in the revised manuscript. The corrections are highlighted in blue color in the revised manuscript.
- Line 116: Take care of the numbers. There are seven numbers listed when there are only six different categories.
Reply: We thank the reviewer for this comment. As suggested, we have checked and modified this part in the revised manuscript.
- Line 171: Take care of Table number
Reply: Thanks for this comment. The table number is added in the revised manuscript.
- Line 178: It can also be observed …
Reply: We have corrected this grammatical mistake in the revised manuscript.